# Genetic and Pathophysiological Basis of Cardiac and Skeletal Muscle Laminopathies

**DOI:** 10.3390/genes15081095

**Published:** 2024-08-20

**Authors:** Shruti Bhide, Sahaana Chandran, Namakkal S. Rajasekaran, Girish C. Melkani

**Affiliations:** 1Department of Biology, Molecular Biology Institute, San Diego State University, San Diego, CA 92182, USA; shrutivbhide@gmail.com (S.B.); sahaanarc@gmail.com (S.C.); 2Department of Pathology, Division of Molecular and Cellular Pathology, Heersink School of Medicine, University of Alabama, Birmingham, AL 35294, USA; rajnsr@uabmc.edu

**Keywords:** laminopathies, redox-homeostasis, Nrf2-signaling, autophagy-signaling, aging, cardiomyopathy and skeletal muscle dysfunction

## Abstract

Nuclear lamins, a type V intermediate filament, are crucial components of the nuclear envelope’s inner layer, maintaining nuclear integrity and mediating interactions between the nucleus and cytoplasm. Research on human iPSC-derived cells and animal models has demonstrated the importance of lamins in cardiac and skeletal muscle development and function. Mutations in lamins result in laminopathies, a group of diseases including muscular dystrophies, Hutchison–Gilford progeria syndrome, and cardiomyopathies with conduction defects. These conditions have been linked to disrupted autophagy, mTOR, Nrf2-Keap, and proteostasis signaling pathways, indicating complex interactions between the nucleus and cytoplasm. Despite progress in understanding these pathways, many questions remain about the mechanisms driving lamin-induced pathologies, leading to limited therapeutic options. This review examines the current literature on dysregulated pathways in cardiac and skeletal muscle laminopathies and explores potential therapeutic strategies for these conditions.

## 1. Introduction

### Intermediate Filaments, Lamins Types, and Function

The cytoskeleton is the component of the cell responsible for maintaining the shape of the cell and the transportation of nutrients. The cytoskeleton in the eukaryotic cell consists of three major components, named intermediate filaments, microtubules, and micro-filaments [1,2]. Among them, intermediate filaments are more diverse and further classified into six types depending on their similarity in amino acid sequences and protein structure [3]. Nuclear lamins belong to type V intermediate filaments and are major components of the inner layer of the nuclear envelope rather than the cytoskeleton, making this type unique in function. Nuclear lamins and associated membrane proteins form a proteinaceous layer between the inner nuclear membrane and chromatin [2]. Some of the major proteins associated with the nuclear membrane exist in the Nuclear Pore Complex (NPC) and LINC complex. The NPC provides a passage for signaling protein and transcription factors between the nucleus and cytoplasm [4]. The LINC complex (linker of nucleoskeleton and cytoskeleton) facilitates structural links between the nuclear lamins, inner nuclear membrane, and actin filaments [4].

Invertebrates and lower organisms consist of only one B-type lamin, except for Drosophila, which contains both lamin-A (referred to as laminC or LamC) and lamin-B (referred to as lamin Dm0). Vertebrates have three lamin genes (one lamin A and two lamin B), with Xenopus having three lamin B genes as an exception [5,6]. Based on previous studies, mammals primarily have two main lamin types, which are referred to as A- and B-type [1,2]. B-type lamins encode for LMNB1 and LMNB2, which are ubiquitously expressed during early development stages including neuronal development [7,8,9]. A-type lamins encode for the LMNA gene which has four isoforms (lamin-A, lamin-C, AΔ10, and C2) as a result of alternative splicing in humans [1,2,10,11,12]. Lamins A/C have been primarily linked with regulating gene expression and myocyte differentiation [4,13]. However, lamin C2 serves as a key structural protein during mitotic processes [14,15]. The structure of lamins consists of a 664 amino-acid-long chain which is divided into three main domains. The N-terminal domain, the head domain, is the smallest among the three domains and is common among all the lamins. The central coil–coil rod domain is the largest domain, consisting of an α-helical structure. The C-terminal is the Ig fold domain, also known as the tail domain [1,10,11,12,16]. The Ig fold domain distinguishes lamin-A from lamin-C [16] due to the C-terminal tail domain region. Lamin-A is derived from its precursor, known as prelamin-A [1,2,9]. Lamin-C lacks structural regions encoded by two exons as compared to prelamin-A. A CAAX region is present in prelamin-A [9]. This region undergoes various post-translation modifications including the farnesylation of Prelamin-A, which is vital to form mature lamin-A. On the contrary, the lamin-C isoform lacks a post-translational modifications process [1,2,9]. The expression levels of lamin-A and its variant, lamin-C, the two primary isoforms encoded by the LMNA gene, vary according to the cell type and tissue [17]. These variations are believed to influence the mechanical stability of cell nuclei. Cells in tissues that are structurally rigid or experience high shearing stresses, such as bone, muscle, and skin, typically exhibit high levels of lamin-A and -C, resulting in stiffer nuclei. Conversely, cells in less dense tissues, especially those in the hematopoietic system, generally show low or undetectable levels of lamin-A and -C, leading to more flexible nuclei. This includes inactive T and B cells as well as differentiated neutrophils [18].

Nuclear lamins surround the nuclear envelope and thus serve as key components in maintaining the structure and shape of the nucleus [11,19,20]. Previous studies have shown that mutation in *Drosophila* lamin Dm0 inhibits the assembly of nuclear membranes and forms lamellae annulate [2,21]. Furthermore, it affects nuclear morphology, growth, and development in Drosophila [22]. Lamin-A function is also linked with chromatin positioning, gene expression, nuclear assembly during mitosis, the replication of DNA, repair pathways, and the regulation of transcription factors [11,17,18]. A lack of lamins leads to the inhibition of chromatin association as well as the improper formation of nuclear membrane and assembly in Xenopus [23]. Studies conducted in *Caenorhabditis elegans* show that interference in lamin expression leads to structural changes in nuclear shape, the organization of the nuclear pore complex, and improper chromosome segregation [24]. Mouse models with a homozygous *LMNA* nonsense mutation showed lethal phenotypes [24,25]. Mutations in lamin also lead to cardiac and skeletal muscle defects [26,27,28]. Apart from the above, lamin functions also consist of the proliferation of cells as well as G1/S phase transition [11,19,20]. Since lamins consist of the inner layer of the nuclear envelope, their role has also involved the facilitation of interactions with the cytoskeleton through Sad1p, UNC-84 (SUN), and nesprin proteins, which are part of LINC complexes [11,19,20,29,30,31]. Lamin is a key component in the transportation of molecular components between the nucleus and cytoplasm via nuclear pore complexes [24,32]. The effect of lamin mutations on nuclear stability, lamin assembly, and nucleo-cytoskeletal coupling have been reviewed in detail by Davidson and Lammerding [33]. 

## 2. Laminopathies (One Gene Multiple Pathologies)

There are multiple mutations linked with the lamin (LMNA) gene that lead to various diseases commonly referred to as laminopathies. There are over 400 mutations resulting in multiple diseases [34,35,36,37]. Some of these diseases include Emery–Dreifuss muscular dystrophy (EDMD), limb–girdle muscular dystrophy (LGMD), congenital muscular dystrophy (CMD), dilated cardiomyopathy (DCM), and Hutchinson–Gilford progeria syndrome (HGPS) [34,35,36,37]. Laminopathies are classified as rare diseases, and the prevalence varies based on the phenotypes. FLDP cases until 2017 were 1.7–2.8 per million [38] severe cases of FLDP also share features with metabolic disorders [39]. Laminopathies are still complex to understand and can be divided into tissue-specific and systemic laminopathies. The first category includes tissues affected in an isolated manner such as striated muscle, peripheral muscle, and adipose tissue, e.g., lipodystrophy and familial partial lipodystrophy of the Dunnigan type (FPLD), which is caused due to a heterozygous mutation LMNA gene [27,28,40,41]. The second category comprises affected tissues in a systematic manner leading to premature aging and related disorders. Although [41] there have been over 400 mutations characterized in this lamin gene, these mutations do not correlate with the domain and are distributed throughout the lamin gene [10,42]. Patients suffering from laminopathies show phenotypes such as muscular and cardiac dystrophy, progeria, diabetes, dysplasia, lipodystrophy, and neuropathies [27,28,41]. These patients have a very short lifespan, and the major cause of death is cardiomyopathy [10,27,28,42].

## 3. Striated Muscle Laminopathy

About 70% of pathologies linked with the mutations of lamins are known to affect the skeletal muscles. Skeletal muscle laminopathies include diseases, which include dystrophies such as EDMD, LGMD, and CMD [43,44,45] (Figure 1A and Table 1). The diseases can be transmitted by dominant inheritance [43,44,45]. CMD is most frequently present at birth or typically shows a phenotype within the first 24 months. Two different types of phenotypes are seen to be associated with this disease. The first phenotype is more severe and leads to the absence of motor development, whereas the second is milder, wherein the patients are ambulatory but have dropped head syndrome [46]. CMD patients mostly suffer from respiratory failure and cardiac impairment. Major symptoms associated with mutant LMNA-linked skeletal myopathies include weakness of muscle, possibly by muscle wasting, and the tightening of muscles and elbow joints, as seen in EDMD or LGMD [41]. The symptoms are usually preceded by heart defects including conduction defects, dilated cardiomyopathy, and dysrhythmias. Later on, in the majority of laminopathy patients, uncontrolled cardiac dysrhythmias are the major reason for death [41,47,48,49].

Cardiomyopathies, along with conduction defects, are diseases related to a weakening of the heart muscle and the failure of impulse transmission, leading to compromised heart function [50]. Cardiomyopathies are categorized into three types based on tissue phenotypes [49]. Hypertrophic cardiomyopathies (HCM) are caused due to thickening and an increase in the left ventricular size, resulting in stiff ventricles [51]. Dilated cardiomyopathies (DCM) result from weak heart muscles that are unable to contract after diastole [51]. Restrictive cardiomyopathies (RCM) are where the heart is unable to relax its muscle after systole, which results in an enlarged atrium [51,52]. Nearly 64% of all *LMNA* mutation patients suffer from DCM with conduction defects [51,52,53]. Moreover, a few studies revealed that dilated cardiomyopathy patients who are at higher risk for sudden cardiac death are associated with uncontrolled cardiac dysrhythmias [54]. There has been evidence suggesting that mutations in lamins lead to cardiac disorders, resulting in reduced cardiac performance progressing with age [55,56]. Studies using the *Drosophila* model have shown a mutation-specific decrease in lifespan; however, the overexpression of autophagy-related 1 (ATG-1) led to an increase in lifespan [57]. Studies on knockout mouse models (*Lmna*–/–) have shown the development of DCM and reduced lifespan [28]. Heterozygous models of mice have also been shown to have conduction system diseases progressing with age and decreased lifespan [58]. LMNA gene mutations encoded for A/C-type lamins have been linked with accelerated aging [59,60,61]. Studies have also shown that cardiac diseases with severe disease phenotypes such as arrhythmias and heart failure progress with age [61].

**Figure 1 genes-15-01095-f001:**
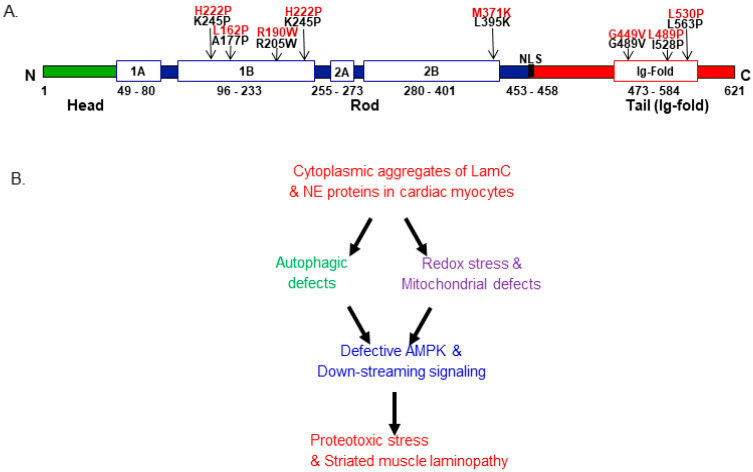
Lamin structure with mutations linked to laminopathies and crosstalk among signaling pathways. (**A**) Lamin structure and the location of mutated amino acids. Black: Drosophila (LamC) and red (LMNA) corresponding human numbering. The Drosophila LamC was used to depict the lamin structure with various domains. (**B**) The crosstalk between autophagy and redox signaling is linked with striated muscle laminopathies. The cytoplasmic aggregation of mutant LamC and other nuclear envelope protein (NEP) impairment triggers cellular and molecular stress, which leads to defective autophagy and impaired redox signaling. Impairment associated with autophagy and redox signaling leads to the inactivation of AMPK and downstream pathways, which produces proteostasis stress and leads to striated muscle laminopathy [57,62].

## 4. Modeling Striated Muscle Laminopathies

Cells obtained from laminopathy patients have been used to study the pathogenic mechanism of striated muscle laminopathy. A rare nonsense p.Y259X homozygous mutation was detected in a deceased newborn, known to cause a complete absence of A-type lamins when present in a homozygous state [63]. The grandmother also carried this mutation [63]. While the cells exhibited abnormal nuclei and mislocalization of lamin B1 and B2, the skin fibroblasts with the same mutation showed no deformities [63]. With the advances in technology, iPSCs were made to recreate the pathology more accurately as well as correct the mutation using gene editing, which was a challenge due to species variance [64,65,66,67].

The worm *(C*. *elegans)* exhibits only a single gene for lamins, known as *lmn-l*, and encodes for lamin-1. This is functionally like both lamin-A and lamin-B found in humans and is also associated with lamin disassembly and nuclear aggregates [68]. Some mutations were lethal, as previously shown for the fourteen laminopathic mutations [69]. In another study, *lmn-l* was knocked down in the gonads and it resulted in abnormal embryos with abnormal nuclei and chromatin. The few animals that survived had very few germ cells and were sterile [68]. *Danio. rerio* has a lamin gene, *lmna*, which is the human orthologue of the human *LMNA* gene. In a study, the zebrafish *lmna* was knocked down, which resulted in skeletal muscle disassembly 24 h post-fertilization [70]. There are two lamin genes encoded by the *Drosophila* genome: *lamin Dmo*, which is like the B-type lamin found in humans, as it is expressed throughout the development of the organism and has the CaaX box in the tail domain; and *lamin-C (LamC)*, which, in *Drosophila*, encodes for the human A-lamin type and is like the *LMNA* gene. The A-lamin type in *Drosophila* lacks the specific sequence called CaaXbox. Moreover, one of the striking similarities between *Drosophila LamC* and the human *LMNA* gene is that they both share the same intron and exon positions [71].

Several studies have been conducted to study laminopathy caused due to mutant *LamC* in *Drosophila*. *Dm* knockout in *Drosophila* leads to lethality due to changes in the transcriptional activation of genes required in heterochromatin organization in somatic tissues, showing the role of *lamin Dm* in chromatin maintenance [22,72,73]. Neuroblasts in which *lamin Dm* has been knocked out showed hypertrophy in the central nervous system tissues and gonads [72,74]. To investigate the function of *LamC* in Drosophila, siRNA *LamC* knockdown was carried out at the developmental stages. The results showed the role of *LamC* in chromatin organization, and hence its requirement for fly development [75]. Also, the loss of *LamC* resulted in muscle atrophy, a phenotype that is observed in human laminopathy patients. This data shows that lamin-C is structurally and functionally like vertebrate lamin-A/C. Hence, fly models were used to study human laminopathy-causing mutations [72,76]. Nuclear envelope defects were observed in null *LamC* flies, which is a phenotype that is observed in humans with mutations in *LMNA* [66] Human laminopathy-causing mutations were caused in Drosophila *LamC* and the salivary gland and epithelial tissue from different stages of larvae were studied to see the effects. Nuclear lamin aggregates in O-ring structures and nuclear defects were observed [66]. In another study, the expression of human laminopathy mutations in the larval muscles in *Drosophila* led to the severe disorganization of myofibrils, abnormally shaped and spaced nuclei, as well as the cytoplasmic aggregation of nuclear pore proteins and mutant LamC [77,78]. In another study, autosomal dominant mutation LMNA-*E159K*, which causes HGPS in humans, was made in the Drosophila *LamC* gene [79]. It is the dominant human disease-causing mutation. When compared to age-matched wild-type flies, the heterozygous expression of *LamC-E174K* resulted in swollen mitochondria and myofibrillar disorganization at 3 weeks of age. The *LamC-E174K* flies showed defective proteostasis and the accumulation of reactive oxygen species, which are hallmarks of progeria that are observed in human patients [79]. Overall, the findings from these studies revealed that the expression of mutant LamC in *Drosophila* causes muscle, nuclear, and mitochondrial phenotypes that are like the ones that are observed in humans.

A mouse model with disrupted *Lmnb1* and/or *Lmnb2* was shown to have developmental defects in the brain resulting in a reduced number of neurons, abnormalities in nuclear shape in neurons, and premature aging [80]. However, deletions of B-lamin, specifically in the skin and hair, did not lead to abnormal phenotypes, thus suggesting that B-lamins may not be required for basic cellular functions [81]. Additionally, mouse models have been generated for the pathophysiological basis of the diseases [82,83,84,85]. For example, *LMNA*−/− (null) mice models have been shown to mislocalize Emerin to the cytoplasm, a common phenotype seen in EDMD26. LmnaH222P/H222P have shown reduced cardiac and skeletal muscle function, resulting in DCM, reduced locomotion, and premature aging [28,82]. The LmnaN195K/N195K mice model showed the mislocation of transcription factor Hf1b/Sp4, loss of sarcomere function, and reduced lifespan to arrhythmia [20]. Several studies have been carried out using different animal models to study the various point mutations that are known to cause striated muscle laminopathy. Different point mutations causing striated muscle laminopathy have been studied in mouse models [20,86]. In one of these studies, *LMNA* mutation *H222P* was generated using mouse models, which has been previously linked with autosomal dominant EDMD in humans. The homozygous mutant line in mice showed phenotypes like those that were observed in humans, including growth retardation, locomotion, and conduction defects, and died between 9 and 13 months of age. Histopathological analysis of the striated muscle (both cardiac and skeletal) revealed increased fibrosis as well as muscle degeneration. However, heterozygous mice with the same mutation did not show any phenotypes, although, in humans, EDMD is mostly caused by an LMNA autosomal dominant mutation [20,86]. EDMD was studied in mice with another human mutation, *M371K.* The mutant *lmna* was expressed specifically in the heart. Histological analyses of the heart muscles showed nuclear defects and abnormal pathology. The number of mice born with the mutation was lower than expected and the lifespan of the mice was between 2 and 7 weeks, while the wild-type mice lived for 24 months [82,86]. In another study, *LMNA N195K* missense mutation was studied in a mouse model. This LMNA mutation resulted in DCM in humans. The homozygous mutant mice expressing the mutant protein showed similar phenotypes to those seen in humans such as dilated hearts, fibrosis of the cardiac muscles, conduction defects, nuclear shape defects, and age-dependent progression of the disease. The mutant mice all died within an average of 3 months [20,82,86]. HGPS also has been studied in a mouse by modeling the *LMNA-L530P* mutation, which is associated with EDMD in humans. The homozygous mutant mice had significantly compromised growth after 4 days of birth and all the mice were not viable after a month. The mice also showed immobility in joints and pathologies observed in muscle were like human progeria patients [87]. However, most of the studies only report the defects and symptoms associated with the expression of the homozygous point mutations, as no defects were seen in heterozygous mutant animals [20,82,86]. In humans, however, most of the diseases are caused due to autosomal dominant effects. In this thesis, the dominant effects of expressing *LamC* mutants in *Drosophila* are studied, which would more closely recapitulate the dominant mutant effects that are observed in humans.

## 5. Pathways and Signaling Associated with Striated Muscle Laminopathies and Aging

Previous studies have shown that patients suffering from laminopathies have a shorter lifespan, and cardiomyopathy is the major cause of death [50,55,88,89] (Table 1). Despite this, the interaction between the lamin gene and heart function is not well understood. Past findings have demonstrated the role of various genes/pathways including the mammalian target of rapamycin (mTOR); nuclear erythroid-2-p45-related factor-2 (Nrf2), known as CncC in Drosophila; and autophagy-related genes, which are affected due to mutations in lamin in skeletal muscle [78,90,91,92]. A *Drosophila* model mimicking the same human mutation showed cardiac dysfunction progressing with age. Mutant flies were shown to have reduced heart contractility progressing with age [57]. The transmission electron microscope (TEM) images of 3-week-old mutant flies showed severe myofibril degeneration and disorganized Z disks [57]. The study further showed increased nuclear blebbing and myofibril disorganization in mutant flies (*LamC-G489V* and *R205W*).

## 6. Redox Signaling and Autophagy Association with Laminopathies

To understand the pathophysiological basis of mutant *LamC*-induced cardiac muscle and skeletal muscle abnormalities, a *Drosophila* model was generated. The effect of mutant *LamC* on autophagy and redox signaling in the *Drosophila* model was also examined. The authors’ findings revealed that the level of Drosophila *refractory to sigma P* (Ref(2)P), the homolog of mammalian polyubiquitin binding protein p62, was significantly enhanced in both cardiac and skeletal muscle [57,62,78]. Previous findings revealed the association of p-62 with autophagy and the dysregulation of Nrf2-Kelch-like ECH-associated protein 1(Keap1), which leads to impairment of the redox pathway and the progression of cardiac and muscle phenotypes [57,78,93,94]. P62 does have many binding regions (known as an adaptor protein), therefore p62 is involved in the binding of misfolded/aggregated proteins and facilitates their degradation via autophagy and the proteasome [95,96]. Nonetheless, the mechanistic basis by which p62 affects autophagic dysregulation with laminopathy still needs to be explored. Findings using mice models have revealed that a lack of A-type lamins leads to heart and skeletal muscle abnormalities, which is primarily attributed to mTOR, thus impacting the autophagy rate [27,97]. It is also shown that autophagy is vital for modulating levels of B1-type lamin [98]. It is likely possible that aggregates of nuclear envelope proteins (NEP) linked with striated muscle laminopathy could negatively affect autophagy flux [57,62,78,79,93,94]. The hindering of autophagic flux light is associated with defective mitochondria, which eventually results in the upregulation of the TOR pathway (Figure 1). Using *Drosophila* models of striated muscle laminopathies, the aggregation of cytoplasmic NEP resulted in an elevated level of Ref(2)P/p62 and influenced Nrf2-keep signaling by binding with Keap [57,77,93,94]. To summarize, laminopathy directly leads to the upregulation of p62, followed by the upregulation of the TOR pathway, thus hindering autophagy in the striated muscle [99] (Figure 1).

## 7. Proteostasis Signaling with Cytoplasmic-Linked Aggregation Laminopathies

Using the *Drosophila* skeletal muscle laminopathy model, it has been established that the overexpression of AMP-activated protein kinase (AMPK) ameliorates mutant phenotypes linked with mutant *LamC* [62]. More specifically, the overexpression of AMPK and the modulation of downstream signaling can suppress mutant LamC-induced aggregation and dysfunction. Moreover, the modulation of this signaling can sustain homeostasis autophagy and mitochondria, which was compromised under skeletal muscle laminopathy [62]. Despite other functions, AMPK serves as a cellular energy and metabolic sensor, involved with the regulation of autophagy, proteostasis, and mitochondrial homeostasis [100]. We hypothesize that mutations in LamC and NE proteins lead to their aggregation in the cytoplasm of cardiac myocytes. This causes proteotoxic stress, leading to structural and functional changes in cardiac myocytes, resulting in cardiac muscle laminopathies. We also hypothesize that AMPK and downstream signaling, through their interactions with autophagy, protein folding, and redox stress pathways, suppress protein aggregation and maintain cellular homeostasis. Furthermore, the impairment associated with autophagy and redox signaling leads to the inactivation of AMPK and downstream pathways, which imbalance proteostasis signaling and lead to striated muscle laminopathy. AMPK exists as heterotrimeric complexes made of catalytic α subunits and regulatory β and γ subunits and has conserved function across the species. Using mouse aging models, it has been established that enhanced levels of AMPK can ameliorate several aging dysregulations including mitochondrial dysfunction and weight gain [101]. More interestingly, metformin, which serves as an activator of AMPK, downregulates progerin levels (a precursor of laminopathy) as well as defects linked with an HGPS-induced pluripotent stem cell model [102]. In most of these findings, the overexpression/activation of AMPK acts through AMPK downstream signaling including peroxisome proliferator-activated receptor-γ coactivator *(PGC)1α* and Forkhead box transcription factors of the class O *(Foxo)*, by maintaining both metabolic and cellular homeostasis (aggregates, achieving autophagic and mitochondrial homeostasis) [62]. Additionally, Foxo-4E-binding protein (4E-BP) signaling, which is downstream of AMPK, is known to modulate age-linked proteostasis, including the mitigation of aggregation, in the skeletal muscle linked with mutant LamC [103]. As shown, an enhanced level of *4E-BP*, a key downstream of mTOR [104], possibly suppresses TOR activity and ameliorates phenotypes in mouse models with rapamycin treatment [27,97,103]. More specifically, *4E-BP* expression in the muscle suppressed several defects including protein aggregation and metabolic dysregulation using in vivo models of *Drosophila* and mice [62,103,104]. On the contrary, the ubiquitous expression of *4E-BP1* reduced the lifespan of *Lmna*−/− mice, likely by the activation of lipolysis [105]. Moreover, in the *Drosophila* skeletal muscle laminopathy models, increasing the *S6K* level further deteriorates muscle function, and the overexpression of the dominant negative version of *S6K* ameliorates muscle dysfunction. Overall, as indicated in the revised review, AMPK and downstream signaling apparently modulate autophagy, which is responsible for the suppression of the cytoplasmic aggregation of the nuclear pore protein (NPP) linked with *LamC* mutation [62].

Furthermore, the enhanced level of Ref(2)P/p62 is associated with the activation of mTOR complex 1 (*RPTOR*), through binding with mTOR and the inhibition of autophagy [106]. Thus, there are two potential mechanisms for the downregulation of autophagy associated with compromising proteostasis. To further these results, the activation of mTOR signaling causes the enhanced activity of downstream signaling, including S6K, which results in the dysregulation of the homeostasis process [107,108]. The RNA sequencing data acquired from muscle biopsy tissue carrying *LMNA-G449V* mutation showed the upregulation of RPTOR and S6K, suggesting potentially compromising autophagy [62] (Figure 2). Using the grain-of-function and loss-of-function approach has been shown to modulate mutant LamC, as established in the *Drosophila* laminopathy skeletal muscle model [88]. Moreover, the suppression of autophagy is anticipated to compromise AMPK activity [104]. To support this prediction, *AMPKα* levels were found to be reduced in muscle biopsy tissue carrying the *LMNA-G449V* mutation, and the muscle-specific over-expression of *AMPKα* was sufficient to rescue muscle C-induced muscle dysfunction and aggregation using *Drosophila* models [62] (Figure 2). Moreover, it has been shown that the inactivation of AMPK resulted in the inactivation of Phosphoinositide 3-kinases/AKT Serine/Threonine Kinase 1 (PI3K/AKT) and mTOR signaling. AMPK’s additional vital function involves controlling the expression of several genes linked with energy metabolism and aging, possibly by boosting the activity of sirtuin 1 (SIRT1) [109,110]. It has also been shown that SIRT1 controls the activity of downstream targets like PGC-1α, which is a central regulator of mitochondrial biogenesis, as well as Foxo, a key gene linked with delaying the aging process [111,112,113]. Additionally, RNA sequencing data acquired from muscle biopsy tissue carrying the *LMNA-G449V* mutation showed the upregulation of several kinases [88], suggesting their activation or activation with laminopathy (Figure 2), including striated muscle dysfunction [63,112].

## 8. Treatments and Future Prospective

Using genetic engineering approaches, researchers have shown the modulation of several genes/pathways including AMPK and downstream signaling, which are vital for autophagy, energy metabolism, redox signaling, and the aging process. This genetic manipulation led to the identification of vital key players in the amelioration of striated muscle laminopathy [62]. Indeed, the identified genes/pathways in these findings are valuable pathogenic markers for skeletal muscle laminopathy and mitochondrial dysmetabolism. The results from these pathway analyses offer promising directions for the next generation of rational therapeutic strategies. Furthermore, by understanding the interplay between these key players, including autophagy, AMPK, and redox pathways, researchers can explore innovative approaches to combat striated muscle laminopathy and other laminopathy disorders. In vivo, models like Drosophila can be used for the genetic modulation of the above-mentioned pathways, including kinases, and the interplay among various potential players in mitigating striated muscle laminopathy. In addition to in vivo genetic validations, feeding rapamycin (a TOR inhibitor) or 5-Aminoimidazole-4-carboxamide ribonucleoside (AICAR), an activator of AMPK, using *Drosophila* models was able to mitigate mutant LamC-induced striated muscle laminopathy [62]. As tested in the *Drosophila* models, these pharmacological compounds/potential drugs have the potential to be tested in mouse striated muscle laminopathy models [98,119]. The *LMNA* gene is associated with the maximum number of disease-causing mutations identified so far in one human gene, commonly referred to as laminopathies. The mutant lamin-induced pathogenic genes/pathways summarized in this review will be vital for understanding human pathology and will be potential targets for more translational/clinical studies. Moreover, the significance of this review will be helpful to provide awareness into therapies for striated muscle laminopathies, which are linked with severe deadly arrhythmias in humans. Even beyond striated muscle pathologies, the lecture included in this review also might be relevant to suppressing other human laminopathies, such as lamin-linked lipodystrophy and devastating progeria.

## Figures and Tables

**Figure 2 genes-15-01095-f002:**
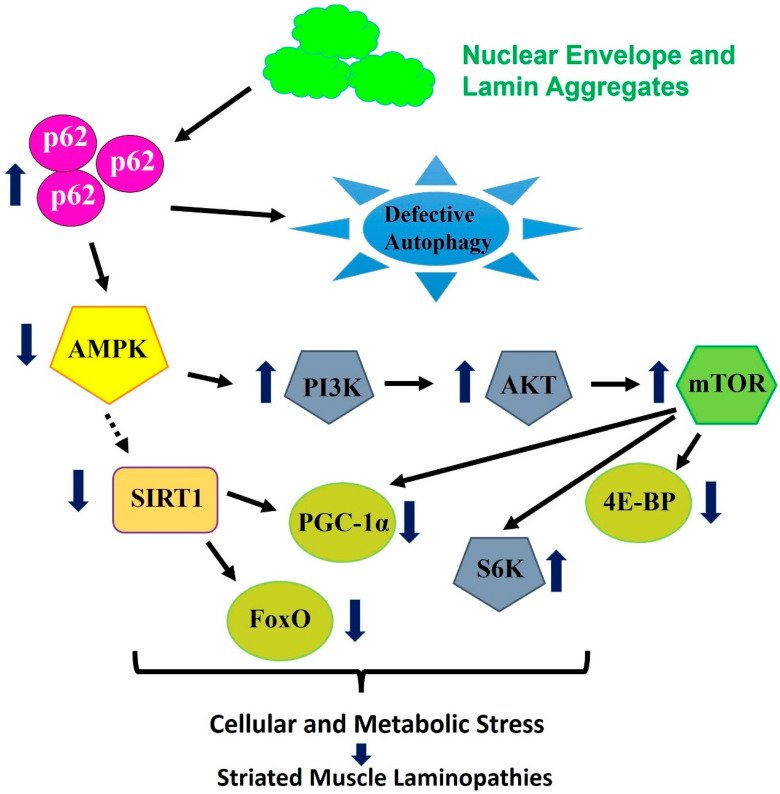
Model for the mechanism of AMPK/TOR/autophagy signaling by cytoplasmic lamin aggregates. Modified from Ref. [62], tissue-specific expression of lamin mutations in the heart and skeletal muscle causes the aggregation of mutant LamC and others (NEP), which causes an upregulation of p62, which ties with the aggregates and directs them for degradation [96]. Abnormal accumulation of p62 leads to autophagy inhibition in the heart and skeletal muscle, which leads to the inactivation of AMPK, which functions to maintain cellular energy homeostasis [114]. The inactivation of AMPK is associated with the hyperactivation of TOR signaling, a pathway associated with cell growth and cell survival [115,116]. In transcriptomics data obtained from muscle biopsy tissue carrying the *LMNA-G449V* mutation, the above-mentioned genes were upregulated, thereby leading to autophagy inhibition in the cardiac muscle [106]. The upregulation of mTOR activity also causes enhanced S6K activity and leads to an imbalance in energy homeostasis [116,117]. Moreover, AMPK is also known to regulate the expression level of genes associated with SIRT1, a class III histone deacetylase, which is linked with proteostasis and energy metabolism [110]. Furthermore, SIRT1, in turn, regulates the activity of downstream signaling, including PGC-1α, which is involved with the biogenesis of mitochondrial and FOXO control proteostasis and target 4E-BP, known to be associated with growth [112,113,118]. In addition to in vivo models, transcriptomics data obtained from human muscle tissue showed alterations in SIRT1 and other genes, thus leading to compromising proteostasis, inducing cellular stress and leading to the impairment of striated muscle pathology [57,62].

**Table 1 genes-15-01095-t001:** Major laminopathy mutations associated with cardiac and skeletal muscle dysfunction.

LMNA/LamC (Human/Drosophila)	Observed Striated Muscle Phenotype/Life Span in Mouse or *Drosophila* Models (Only Human Amino Acids LMNA Mutations Drosophila Orthologs Are Shown)
*Lmna*–/–)	Compromised cardiac and skeletal muscle performance and short lifespan.
Lmna-*E159K*	Swollen mitochondria, defective proteostasis, enhanced ROS, and myofibrillar disorganization.
*Lmna-L162P*	Progressive muscle dysfunction, shuttle nuclear blebbing, and LamC aggregates.
*Lmna-R190W*	Dilated cardiomyopathy, muscle dysfunction, nuclear blebbing, LamC aggregates, short lifespan, defective autophagy, and redox signaling.
*Lmna-N195K*	Mislocation of transcription factor Hf1b/Sp4, loss of sarcomere function, reduced lifespan to arrhythmia.
*Lmna-H222P*	Muscular dystrophies and dilated cardiomyopathy.
*Lmna-M371K*	Nuclear defects, abnormal heart pathology, and short lifespan.
*LamC-G449V*	Constricted heart, muscle dysfunction, nuclear blebbing, cytoplasmic LamC aggregates, short lifespan, defective autophagy, and redox signaling.
*LamC-L489P*	Severe skeletal myopathy with nuclear blebbing, and cytoplasmic LamC aggregates.
*LMNA-L530P*	The homozygous mutant mice have significantly compromised growth and all the mice were not viable after a month.

## Data Availability

Not applicable.

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
