# Peer review of "Genetic and Pathophysiological Basis of Cardiac and Skeletal Muscle Laminopathies"

_genes, 2024, doi:10.3390/genes15081095_

Round 1

Reviewer 1 Report

Comments and Suggestions for Authors

The authors aimed to review nuclear lamin-induced cardiac and skeletal muscle laminopathies as well as potential therapies for lamin-based striated muscle pathologies.

The text was well written.  However, I raised only a comment for general readers.

The reader wants to know the molecular mechanism of how nuclear lamin works in muscle cell and its deficiency causes muscle diseases?  It is difficult to understand how the nucleus affects muscle force development of actomyosin or sarcomere structure or organization? Please explain it using cartoon in the text. Figure 2 is not easy to understand.

Author Response

Reviewer 1:

Comment: The reader wants to know the molecular mechanism of how nuclear lamin works in muscle cell and its deficiency causes muscle diseases?  It is difficult to understand how the nucleus affects muscle force development of actomyosin or sarcomere structure or organization? Please explain it using cartoon in the text. Figure 2 is not easy to understand.

Response: We would like to thank the reviewer for this important question. While we acknowledge that Figure 2 may appear complex, it is essential for illustrating the various signaling pathways, which can be useful for validation purposes in the future. However, we have simplified Figure 2 based on recent evidence. These findings suggest that cytoplasmic aggregates may cause proteotoxic stress, leading to cellular defects that contribute to structural changes in cardiac and skeletal muscle, resulting in striated muscle laminopathies. Furthermore, cytoplasmic aggregation of LMNA may trigger metabolic and cellular responses, including the activation of mTOR and suppression of autophagy and AMPK signaling. As shown in Figure 2 and discussed in the review, AMPK and its downstream signaling, through interactions with autophagy, protein folding, and redox stress pathways, suppresses protein aggregation and maintains cellular homeostasis. Additionally, LMNA mutations may interfere with lamin assembly and compromise nuclear stability, which has also been discussed in the review. This comprehensive review aims to assist readers interested in gaining an in-depth understanding of the mechanistic basis of striated muscle laminopathy.

Figure 2. Model for the mechanism of AMPK/TOR/autophagy signaling by cytoplasmic lamin aggregates. Tissue-specific expression of Lamins mutation in the heart and skeletal muscle causes aggregation of mutant LamC and other (NEP)which causes an upregulation of p62, which ties with the aggregates and directs them for degradation [96]. Abnormal accumulation of p62, leads me autophagy inhibition in the heart and skeletal muscle which leads to the inactivation of AMPK, which functions to maintain cellular energy homeostasis [115]. Inactivation of AMPK is associated with the hyperactivation of TOR signaling, a pathway associated with cell growth and cell survival [116,117]. Transcriptomics data obtained from muscle biopsy tissue carrying LMNA-G449V mutation, the above-mentioned genes were upregulated, thereby leading to autophagy inhibition in the cardiac muscle [106]. Upregulation of mTOR activity also causes an enhanced S6K activity and leads to an imbalance in energy homeostasis [117,118]. Moreover, AMPK is also known to regulate the expression level of genes associated with SIRT1, a class III histone deacetylase, which is linked with proteostasis and energy metabolism [1110]. Furthermore, SIRT1, in turn, regulates the activity of downstream signaling including PGC-1α, which is involved with the biogenesis mitochondrial, and FOXO control proteostasis and target 4E-BP known to be associated with growth [112,119,120]. In addition to in vivo models, transcriptomics data obtained from human muscle tissue showed alterations of SIRT1 and other genes, thus leading to compromising proteostasis, inducing cellular stress, and lead to impairment of striated muscle pathology [57,92].

Reviewer 2:

Minor comments.

  1. 1. Would encourage authors to discuss the LMNA gene a bit more in detail.

What is the size of the gene? What is the expression level across tissues and cell types? What is the expression of lamin isoforms across tissue and cell types especially heart and skeletal muscle?

Response: We appreciate the reviewer's comments. We have added more detailed information about the LMNA gene, including protein structure, domains, and sub-domains, in the second paragraph of the review (lines 51-62), as well as referenced in Figure 1. Additionally, in the same paragraph, we have highlighted the different types of lamins and their tissue-specific expressions. For instance, as mentioned in lines 45-51, B-type lamins are ubiquitously expressed and are associated with regulating gene expression and myocyte differentiation. Furthermore, as noted in lines 61-69, the expression levels of lamin A and its variant lamin C—the two primary isoforms encoded by the LMNA gene—vary depending on cell type and tissue. However, the main focus of this review is on Laminopathies, and several published reviews already cover lamin structure and expression levels across tissues.

  1. On Line 146, authors states " Cells obtained from laminopathy patients..." and followed by "neonatal death" of cells from a grandmother who was heterozygous seems very confusing. Authors should clarify this sentence. What cells were these and what exactly was the genotype? 

Response: We appreciate the reviewer's observation. We have revised the sentence in lines 161-166 to improve clarity. The revised text now reads: "Cells obtained from laminopathy patients have been utilized to investigate the pathogenic mechanism of striated muscle laminopathy. In one case, a rare nonsense p.Y259X homozygous mutation, associated with a complete absence of A-type lamins in the homozygous state, was identified in a deceased newborn [62]. The grandmother also carried this mutation [62]. While the cells exhibited abnormal nuclei and mislocalization of lamin B1 and B2, the skin fibroblasts with the same mutation showed no deformities [62]." Overall, this suggests the disease pathology with Lamin A vs B is different.  

  1. Authors should state the prevalence of Laminopathies.

Response: Laminopathies are rare diseases with a prevalence that varies depending on the severity of the condition. As noted in lines 99-100, the incidence of FLDP cases until 2017 was reported to be 1.7-2.8 per million [38], and severe cases of FLDP also share characteristics with metabolic disorders [39]. Furthermore, as mentioned in lines 101-111, laminopathies can manifest as either tissue-specific or systemic disorders, which may also influence their prevalence. Therefore, additional data is needed to accurately determine the prevalence of laminopathies.

Reviewer 2 Report

Comments and Suggestions for Authors

The authors Bhide at al., in this review summarize the 

1. Laminopathies and the pathophysiology particularly relevant to skeletal muscle and cardiac dysfunction.

2. Discuss relevant animal models of Laminopathies that can be utilized to further dissect the mechanism of pathophysiology.

3. discuss the signaling mechanisms implicated in laminopathy disease state. 

4. Potential therapeutic interventions to treat Laminopathies

Minor comments.

1. Would encourage authors to discuss the LMNA gene a bit more in detail.

What is the size of the gene? What is the expression level across tissues and cell types? What is the expression of lamin isoforms across tissue and cell types especially heart and skeletal muscle?

2.  On Line 146, authors states " Cells obtained from laminopathy patients..." and followed by "neonatal death" of cells from a grandmother who was heterozygous seems very confusing. Authors should clarify this sentence. What cells were these and what exactly was the genotype? 

3. Authors should state the prevalence of Laminopathies.

Author Response

(The authors gave the same response as above.)
